# Efficacy of the CDK7 Inhibitor on EMT-Associated Resistance to 3rd Generation EGFR-TKIs in Non-Small Cell Lung Cancer Cell Lines

**DOI:** 10.3390/cells9122596

**Published:** 2020-12-03

**Authors:** Wonjun Ji, Yun Jung Choi, Myoung-Hee Kang, Ki Jung Sung, Dong Ha Kim, Sangyong Jung, Chang-Min Choi, Jae Cheol Lee, Jin Kyung Rho

**Affiliations:** 1Department of Pulmonology and Critical Care Medicine, Asan Medical Center, College of Medicine, University of Ulsan, Seoul 05505, Korea; jack1097@naver.com (W.J.); ccm9607@gmail.com (C.-M.C.); 2Asan Institute for Life Sciences, Asan Medical Center, College of Medicine, University of Ulsan, Seoul 05505, Korea; choiyj@amc.seoul.kr (Y.J.C.); mhkang1227@gmail.com (M.-H.K.); kijung1019@naver.com (K.J.S.); Kimdongha@amc.seoul.kr (D.H.K.); 3Department of Biomedical Sciences, Asan Medical Center, AMIST, College of Medicine, University of Ulsan, Seoul 05505, Korea; sangyong1215@naver.com; 4Department of Oncology, Asan Medical Center, College of Medicine, University of Ulsan, Seoul 05505, Korea; 5Department of Convergence Medicine, Asan Medical Center, College of Medicine, University of Ulsan, Seoul 05505, Korea

**Keywords:** lung cancer, CDK7, EGFR-TKIs, EMT, THZ1, resistance

## Abstract

Epithelial to mesenchymal transition (EMT) is associated with resistance during EGFR tyrosine kinase inhibitor (EGFR-TKI) therapy. Here, we investigated whether EMT is associated with acquired resistance to 3rd generation EGFR-TKIs, and we explored the effects of cyclin-dependent kinase 7 (CDK7) inhibitors on EMT-mediated EGFR-TKIs resistance in non-small cell lung cancer (NSCLC). We established 3rd generation EGFR-TKI resistant cell lines (H1975/WR and H1975/OR) via repeated exposure to WZ4002 and osimertinib. The two resistant cell lines showed phenotypic changes to a spindle-cell shape, had a reduction of epithelial marker proteins, an induction of vimentin expression, and enhanced cellular mobility. The EMT-related resistant cells had higher sensitivity to THZ1 than the parental cells, although THZ1 treatment did not inhibit EGFR activity. This phenomenon was also observed in TGF-β1 induced EMT cell lines. THZ1 treatment induced G2/M cell cycle arrest and apoptosis in all of the cell lines. In addition, THZ1 treatment led to drug-tolerant, EMT-related resistant cells, and these THZ1-tolerant cells partially recovered their sensitivity to 3rd generation EGFR-TKIs. Taken together, EMT was associated with acquired resistance to 3rd generation EGFR-TKIs, and CDK7 inhibitors could potentially be used as a therapeutic strategy to overcome EMT associated EGFR-TKI resistance in NSCLC.

## 1. Introduction

Lung cancer is the leading cause of cancer mortality globally [1]. Resistance to lung cancer therapies contributes to the higher death rates of these patients. A targeted therapy utilizing an epidermal growth factor receptor (EGFR) mutation improves the survival rate of patients until resistance begins to develop [2]. First-generation EGFR-tyrosine kinases (TKIs) therapy exhibits efficacy for a short time (<2 years). However, inevitably, resistance gradually develops, leading to disease progression [3]. Various resistance mechanisms for EGFR-TKIs have been reported in previous research, including T790M point mutations, EGFR amplification, ERBB2 amplification, small cell lung cancer transformation, MET amplification, PI3K mutations, epithelial-mesenchymal transition (EMT), BRAF mutations, and KRAS mutations [3,4,5,6]. The T790 mutation is considered the most prevalent type (40~50% of cases) of resistance to the first-generation EGFR-TKIs. Recently, a third-generation drug, osimertinib and WZ4002, has been developed to overcome T790M-associated resistance [7,8]. However, most of the mechanisms involved in third-generation EGFR-TKI resistance have not yet been elucidated. As such, the development of methods to overcome acquired resistance to 3rd generation EGFR inhibitors is critical for improving the prognosis of patients.

EMT is a biological process via which cells undergo a switch from the polarized epithelial phenotype to the mesenchymal fibroblastoid phenotype, and it is considered to be one of the resistance mechanisms. EMT is involved in several diverse processes, including embryonic development, chronic inflammation, fibrosis, tumorigenesis, invasion, metastasis, and drug resistance [9,10,11,12,13]. As a result of EMT, cells downregulate the expression of epithelial proteins such as E-cadherin and upregulate the expression of mesenchymal proteins, including vimentin. In addition, cells undergoing EMT are characterized by the loss of apico-basal polarity and intact cell-cell junctions followed by the acquisition of front-rear polarity and morphologic changes to a spindle shape with remodeling of the cytoskeleton [14]. This EMT program enhances their invasive capacity, therapeutic resistance, and the cancer stem-cell-like properties of the cells [13]. EMT has also been reported to be one of the mechanisms of EGFR-TKI resistance [6]. However, how to overcome EMT related EGFR-TKI resistance is unclear at this time.

Cyclin-dependent kinases (CDKs) are a family of serine-threonine kinases that play an important role in cell cycle progression. Over 90% of tumors have upregulated CDKs due to changes in the expression and genetic variation of CDKIs. Based on this, CDKs inhibitors have potential as anticancer agents that could inhibit the growth of various cancers. In particular, CDK7 acts as a master regulator of transcription and is known to modulate RNA polymerase II activity [15]. Recently, CDK7 inhibitors have been reported to inhibit abnormal cell growth associated with hematologic malignancies and breast, esophageal, and small cell lung cancer [16,17,18,19]. Specifically, small cell lung cancer studies have shown that the action of THZ1 significantly reduces the activity of super-enhancers and their associated oncogene transcription factors [17], which suggests that CDK inhibitors have potential as anticancer agents.

This study evaluated whether EMT was expressed in resistant cell lines generated by treatment with 3rd generation EGFR-TKIs, and we also analyzed the effect of CDK7 inhibitors on the EMT-associated resistant cells to evaluate a potential therapeutic strategy to overcome EMT related EGFR-TKI resistance.

## 2. Materials and Methods

### 2.1. Cell Culture and Reagents

The H1975 and HCC827 cell lines were obtained from the American Type Culture Collection (Rockville, MD, USA), and the PC-9 cell line was a kind gift from Dr. Kazuto Nishio (National Cancer Center Hospital, Tokyo, Japan). The cells were cultured in RPMI 1640 (Invitrogen, Carsbad, CA, USA) that contained 10% fetal bovine serum (FBS), 100 U/mL penicillin, and 100 mg/mL streptomycin (Invitrogen) at 37 ℃ in an atmosphere with 5% CO_2_. Osimertinib, WZ4002, and THZ1 were purchased from Selleck Chemicals (Houston, TX). The 3-(4,5-dimethylthiazo-2-yl)-2,5-diphenyltetrazolium bromide (MTT) solution and TGF-β1 were purchased from Sigma (St. Louis, MO, USA) and R&D Systems (Minneapolis, MN, USA), respectively. QS1189 was kindly provided by Qrient Co., Ltd. (Seongnam, Korea).

### 2.2. Establishment of the 3rd Generation EGFR-TKIs-Resistant Cells

The H1975 harbors the EGFR L858R/T790M double mutation. To establish a cell with acquired resistance to 3rd generation EGFR-TKIs including osimertinib and WZ4002, H1975 cells were initially treated to 10 nM osimertinib or WZ4002 for 48 h. The surviving cells were continuously exposed to increasing concentrations for 3 months, as reported in previous studies [20,21,22,23,24]. The resistant cells (H1975/OR and H1975/WR) were cultured in a drug-free medium for >1 week before the experiment to eliminate the effects of each drug. The resistant cell lines were authenticated using STR analysis and confirmed to be mycoplasma free using standard methods.

### 2.3. Cellular Viability Assays

Cells (1 × 10^4^) were seeded in 96-well sterile plastic plates overnight and then treated with the relevant drugs. After 72 h, 15 μL of MTT solution (5 mg/mL) was added to each well, and the plates were incubated for 4 h. Crystalline formazan was solubilized with 100 μL of 10% (w/v) SDS solution for 24 h. The absorbance at 595 nm was read spectrophotometrically using a microplate reader. Cell counting was determined using an ADAM-MC automatic cell counter (NanoEnTek, Seoul, Korea) according to the manufacturer’s instructions. The results represent at least three independent experiments, and the error bars signify the standard deviation from the mean. The IC_50_ values were determined using GraphPad Prism software (GraphPad, Inc., La Jolla, CA, USA).

### 2.4. Immunoblotting

Whole cell lysates were prepared using EBC lysis buffer (50 mM Tris-HCl [pH, 8.0], 120 mM NaCl, 1% Triton X-100, 1 mM EDTA, 1 mM EGTA, 0.3 mM phenylmethylsulfonyl fluoride, 0.2 mM sodium orthovanadate, 0.5% NP-40, and 5 U/mL aprotinin) and then centrifuged. The resulting supernatants (20 μg) were separated on 8% to 12% sodium dodecyl sulfate polyacrylamide gel electrophoresis gels and then transferred onto polyvinylidene difluoride membranes (Invitrogen). The membranes were probed with antibodies against p-EGFR (1:1000, Tyr1173, sc-101668), EGFR (1:2000, sc-373749), p-Erk (1:1000, Thr202/Tyr204, sc-16982), Erk (1:3000, sc-94), Akt (1:3000, sc-5298), E-cadherin (1:1000, sc-71008), CDK7 (1:2000, sc-7344), EpCAM (1:1000, sc-71059), desmoplakin (1:1000, sc-390975), cytokeratin-8/18 (1:1000, sc-70939) and β-actin (1:5000, sc-47778; all from Santa Cruz Biotechnology, Santa Cruz, CA, USA), p-Akt (1:1000, Ser473, #4060), β-catenin (1:1000, #8480), vimentin (1:1000, #5741), RNAPII CTD p-Ser2 (1:1000, #13499), RNAPII CTD p-Ser5 (1:1000, #13523), RNAPII CTD p-Ser7 (1:1000, #13780), RNAPII CTD (1:2000, #2629), PARP (1:1000, #9541) and caspase 3 (1:1000, #9661; all from Cell Signaling Technology, Beverly, MA, USA) and then the membranes were incubated with a horseradish peroxidase-conjugated secondary antibody. All membranes were developed using ECL kits (Perkin Elmer, Waltham, MA, USA). The immunoblotting is representative of three independent experiments.

### 2.5. Invasion and Migration Assays

The cell migration and invasion assays were conducted using Transwells (6.5 mm diameter, 8 mm pore size polycarbonate membrane), which were obtained from Corning (Cambridge, MA, USA). The cells (1 × 10^5^) in 200 μL medium were placed in the upper chamber, and the lower chamber was filled with 1 mL of serum-free media supplemented with 0.1% bovine serum albumin. After incubation for 24 h, the cells that migrated to the lower surface of the filters were stained with a Diff-Quick kit (Fisher Scientific, Pittsburgh, PA, USA), and then they were counted under a microscope. The invasion assays used the same procedure with filters that were pre-coated with Matrigel (BD Biosciences, Bedford, MA, USA). Triplicate results are expressed as the mean (standard deviation).

### 2.6. Flow Cytometry Analysis

The cells were treated with 0.1 μM THZ1 for the indicated time. To determine their cell cycle process, the cells were fixed in 70% ethanol at −20 °C for 1 h to a few days, incubated with 5 μL RNase (10 mg/mL), and finally stained with 10 μL propidium iodide (1 mg/mL). The cellular DNA content in the treated cells was analyzed with a FACScan flow cytometer (BD Biosciences, Franklin Lakes, NJ, USA). Apoptosis was quantified using an Annexin V-fluorescein isothiocyanate (FITC)/propidium iodide apoptosis kit (BD Biosciences) in accordance with the manufacturer’s protocol. The cells were resuspended in Annexin V-binding buffer (150 mM NaCl, 18 mM CaCl_2_, 10 nM HEPES, 5 mM KCl, and 1 mM MgCl_2_). FITC-conjugated Annexin V (1 μg/mL) and propidium iodide (50 μg/mL) were then added to the cells and incubated for 30 min at room temperature in the dark. Analyses were performed using a FACScan flow cytometer. Data were analyzed with CellQuest software (BD Biosciences).

### 2.7. Statistics

Data are presented as the mean ± standard deviation. *P* values were determined using unpaired or paired *t*-tests between groups using GraphPad Prism software.

## 3. Results

### 3.1. The Induction of EMT in Acquired Resistance to 3rd Generation EGFR-TKIs

To investigate the mechanisms of acquired resistance to 3rd generation EGFR-TKIs, we established H1975/WR and H1975/OR through stepwise selection in WZ4002 or osimertinib, respectively, as described in previous studies [20,21,22,23,24]. Both resistant cell lines exhibited 10-fold or greater resistance against each drug compared with the parent cells (WZ4002 IC_50_ = 106.9 nM in H1975 and 1494 nM in H1975/WR, osimertinib IC_50_ = 39.9 nM in H1975 and 1352 in H1975/OR; Figure 1A). In addition, C797S mutation that can confer resistance to osimertinib was not observed in both resistant cells (data not shown).

These resistant cells showed an increased number of spindle-shaped cells that resembled EMT changes (Figure 1B). To determine the induction of EMT in the resistant cells, we analyzed the expression of marker proteins of the epithelial and mesenchymal phenotypes by using western blots (Figure 1C). Compared with the H1975 cells, the epithelial marker proteins E-cadherin, β-catenin, EpCAM, desmoplakin, and cytokeratin-8/18 were significantly reduced in both resistant cell lines, whereas vimentin expression was increased. In addition, the expression and activity of EGFR were both reduced in the resistant cells, but the activity of Akt was significantly upregulated.

Next, we investigated their migratory and invasive abilities, which are considered functional hallmarks of EMT. We found that the migratory and invasive abilities of the resistant cells were significantly enhanced relative to the parental cells (Figure 1D,E). Taken together, these data suggested that the acquisition of resistance to 3rd generation EGFR-TKIs induced molecular changes that were consistent with EMT.

### 3.2. Efficacy of the CDK7 Inhibitor on EMT-Induced Cells

Previous studies showed that CDK7 was associated with EMT, but it is controversial whether targeting CDK7 can overcome EMT [25,26,27,28]. To determine the effect of CDK7 inhibition on the resistant cells, we used THZ1 and QS1189 as CDK7 inhibitors. QS1189 was developed as a novel CDK7 inhibitor in a previous study [16]. As shown in Figure 2A, both resistant cell lines were more sensitive to CDK7 inhibitors than the parental cells (THZ1 IC_50_ = 379 nM in H1975, 83.4 nM in H1975/WR, 125.9 nM in H1975/OR; QS1189 IC_50_ = 755.3 nM in H1975, 232.8 nM in H1975/WR, 275.3 nM in H1975/OR). CDK7 kinase activity is involved in phosphorylation of the CTD of RNAPII, which plays a role in transcription initiation and RNAPII procession [15,29,30]. To evaluate the inhibitory efficacy of CDK7 substrates on the parental and resistant cells, we performed Western blotting following treatment with THZ1 (Figure 2B). The inhibitory effect of THZ1 on the activity of RNAPII-CTD was similar in the H1975 and H1975/OR cells. However, the H1975/WR cells had inhibition of RNAPII-CTD phosphorylation at Ser2, Ser5, and Ser7 at the lowest concentration of THZ1. In addition, THZ1 treatment did not inhibit the activity of EGFR or Akt, but the activity of Erk showed a dose-dependent induction.

To assess whether the induction of EMT can affect the sensitivity to CDK7 inhibitors, we examined the response to THZ1 under conditions of TGF-β1-induced EMT. In similar to our previous studies [22,31], TGF-β1 treatment led to the induction of EMT through the reduction of E-cadherin and an increase of vimentin (Figure 3A). When the cells were pretreated with TGF-β1 to induce EMT, their sensitivity to THZ1was increased (Figure 3B,C). Consistent with previous studies, the induction of EMT reduced their sensitivity to osimertinib. In addition, TGF-β1 treatment did not affect the inhibition of RNAPII-CTD phosphorylation by THZ1 (Figure 3D). Taken together, the induction of EMT could affect the sensitivity of cells to CDK7 inhibitors.

### 3.3. Induction of Cell Cycle Arrest and Apoptosis by CDK7 Inhibitors

We have previously observed that the mechanisms of the anticancer activity of CDK7 inhibitors were associated with cell cycle arrest and the apoptosis of various cancer cells [16,32,33,34]. Thus, we analyzed the cell cycle and apoptosis. As shown in Figure 4, THZ1 treatment induced G2/M arrest and apoptosis in all cells (Figure 4A,B). Interestingly, the rates of G2/M arrest and apoptosis were higher in both resistant cell lines than in the parental cells (G2/M arrest = 31.6% in H1975, 50.9% in H1975/WR, 30.5% in H1975/OR; apoptosis = 22.7% in H1975, 42.1% in H1975/WR, 33.6% in H1975/OR). Consistent with these results, THZ1 treatment showed an enhancement of cleaved PARP and cleaved caspase 3 (Figure 4C).

### 3.4. Recovery of the Sensitivity to EGFR-TKIs by THZ1-Tolerant Cells

Drug-tolerant (DT) cells develop within several days to several weeks of exposure to specific drugs [35,36]. We generated DT cells from H1975/WR and H1975/OR after 9 days of exposure to 100 nM THZ1. DT cells showed a 3-fold higher IC50 for THZ1 than the parental cells (Appendix A; THZ1 IC_50_ = 72.4 nM in H1975/WR, 200.9 nM in 1975/WR/DT, 124.3 nM in H975/OR, 368.9 nM in H1975/OR/DT). As shown in Figure 5A, these cells showed some morphological changes similar to that seen during mesenchymal-to-epithelial transition (MET). Consistent with these results, the DT cells showed an induction of β-catenin and cytokeratin-8/18, and a reduction of vimentin, although there were no changes in expression of junction proteins, including E-cadherin and EpCAM (Figure 5B, Appendix A). THZ1-DT cells showed a partial recovery of sensitivity to 3rd generation EGFR-TKIs (Figure 5C,D; H1975/WR/DT, IC_50_ = 1.2 μM and H1975/OR/DT, IC_50_ = 1.1 μM; H1975/WR and H1975/OR, IC_50_ > 10 μM). In addition, combined treatment (WZ4002 plus THZ1 in H1975/WR cells, osimertinib plus THZ1 in H1975/OR cells) led to a restoration of sensitivity to 3rd generation EGFR-TKIs in these resistant cells (Figure 5E,F). Taken together, these data suggest that THZ1 treatment could lead to mesenchymal-to-epithelial transition in EMT-induced cells and recover their sensitivity to 3rd generation EGFR-TKIs.

## 4. Discussion

EMT is associated with a poor prognosis of NSCLC patients [37,38] and acquired resistance to various anticancer drugs [39,40,41]. Many studies have shown that EMT affects the sensitivity to EGFR-TKIs, including primary and acquired resistance. Consistent with our studies, recent studies have revealed that EMT is a potential mechanism involved in 3rd generation EGFR-TKI resistance [42]. However, accumulating evidence has revealed that how EMT contributes to EGFR-TKIs resistance is complex and versatile. Previously, some papers demonstrated that TGF-β, IGF1R, AXL, Notch signaling, and miRNA regulating EMT-related molecules are all inducers of EMT in the development of EGFR-TKIs resistance. Although many papers have shown that the inhibition of EMT inducing molecules could overcome the acquired resistance to EGFR-TKIs, they should be studied further to optimize their efficacy in clinical practice.

CDK7 is an essential component of the transcription factor TFIIH, and it facilitates transcription initiation by phosphorylating the serine residues of RNAPII CTD (the C-terminal domain) [43,44]. Thus, CDK7 is an attractive target for anticancer treatment. Inhibition of CDK7 activity inhibits both transcription and cell cycle progression, resulting in the downregulation of anti-apoptotic proteins, such as Mcl-1 and XIAP, and cell cycle regulators, such as cyclin D1 [45]. We also showed that CDK7 inhibition led to the enhancement of G2/M arrest and apoptosis in all cell lines that were used. Although the mechanisms of anticancer effects via CDK7 inhibition have been associated with cell cycle arrest and apoptosis in various cancers [16,46,47], until recently, biomarkers for determining the sensitivity of cells to CDK7 inhibitors have been unclear.

A few limitations of this study should be taken into consideration. Firstly, while our study found that EMT was correlated with acquired resistance to EGFR-TKI, the mechanisms are not yet clear. Although KEGG pathway enrichment analysis revealed that genes involved in EMT-related pathways were enriched in 3rd generation EGFR-TKIs resistant cells (Appendix A), further research is required to identify the specific mechanisms of action involved. Secondly, the detailed mechanisms as to why the EMT induced EGFR-TKI resistant cell lines were more sensitive to CDK7 inhibitors were not investigated. Some papers have suggested that sensitivity to CDK7 inhibitors is associated with factors such as super-enhancer-associated genes, such as SOX2, MYC, and CITED2 [48,49,50]. Among these, MYC and CITED2 were previously described as possible inducers of EMT [51,52], and their expression is enhanced in resistant cells (Appendix A). Additional studies are needed to determine whether the induction of MYC or CITED2 expression in our experimental system can affect EMT and/or the sensitivity to CDK7 inhibitors.

Our results suggest that CDK7 inhibitors may be an attractive treatment option for overcoming EMT-associated EGFR-TKIs resistance. THZ1 treatment led to a reversal of the process of epithelial-to-mesenchymal transition. These phenomena resulted in the recovery of their sensitivity to 3rd generation EGFR-TKIs. Although THZ1 treatment can induce mesenchymal-to-epithelial transition (MET), we think that short-term exposure to THZ1 may help in the clearance of mesenchymal cells because mesenchymal cells are far more sensitive to THZ1 than epithelial cells. Thus, THZ1-tolerant cells can have similar characteristics to the parental cells in our experimental model. Additionally, a combined treatment of 3rd generation EGFR-TKIs and THZ1 was effective against the EMT-induced resistant cells. Our results may suggest various clinical applications to overcome EMT associated EGFR-TKIs resistance due to tumor heterogeneity.

## 5. Conclusions

EMT was associated with decreased sensitivity to 3rd generation EGFR-TKIs, and the EMT-related resistant cells were more sensitive to CDK7 inhibitors. This suggests the possibility of using CKD7 inhibitors as a therapeutic strategy to overcome EMT associated EGFR-TKI resistance in NSCLC.

## Figures and Tables

**Figure 1 cells-09-02596-f001:**
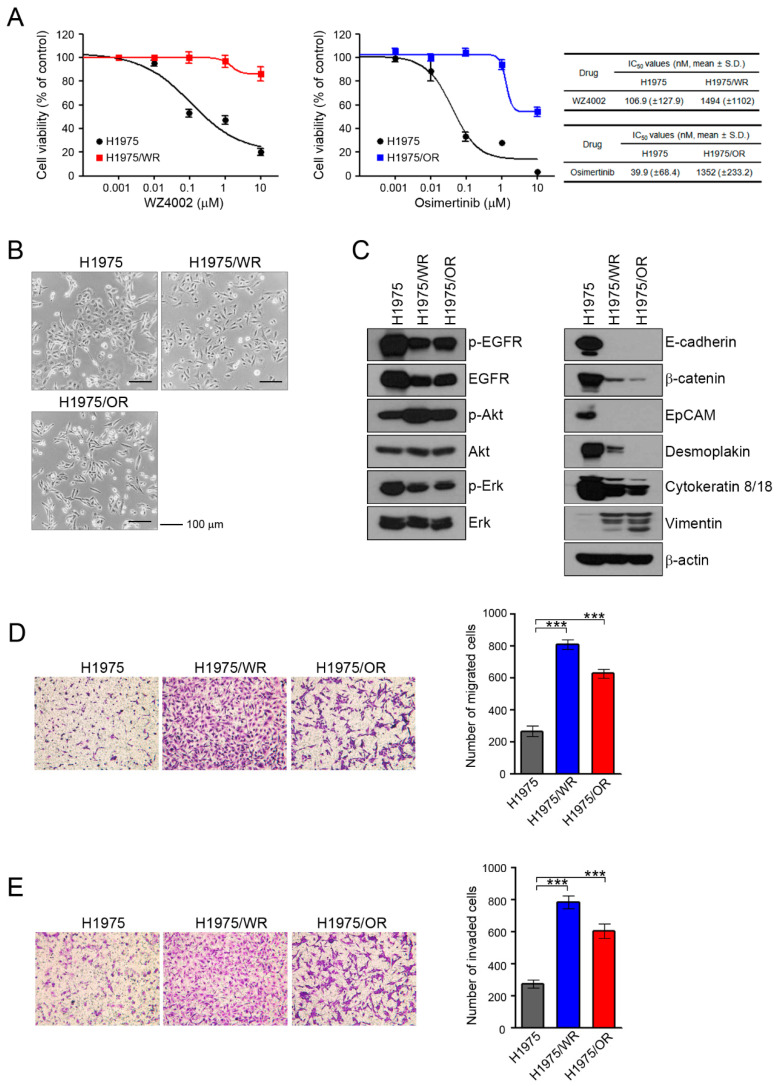
Induction of EMT in cells with acquired resistance to WZ4002 or osimertinib. (**A**) Cells were treated with the indicated doses of WZ4002 or osimertinib for 72 h, and cell viability was determined by an MTT assay. IC_50_ values were calculated with GraphPad software through three independent experiments. (**B**) H1975 parental cells and the resistant cell lines (H1975/WR and H1975/OR) were evaluated for morphologic changes that were consistent with EMT using a light microscope. (**C**) EGFR signaling and EMT-related molecules were analyzed by western blotting. (**D**,**E**) Cells were seeded onto either collagen or Matrigel-coated polycarbonate filters to determine their migratory and invasive potentials, respectively. Cells were incubated in modified Boyden chambers for 24 h, and the cells that penetrated the filter were stained and counted using a light microscope. Experiments were conducted in triplicate. The bars represent the standard deviations. *** *P* < 0.0005 compared with H1975 cells.

**Figure 2 cells-09-02596-f002:**
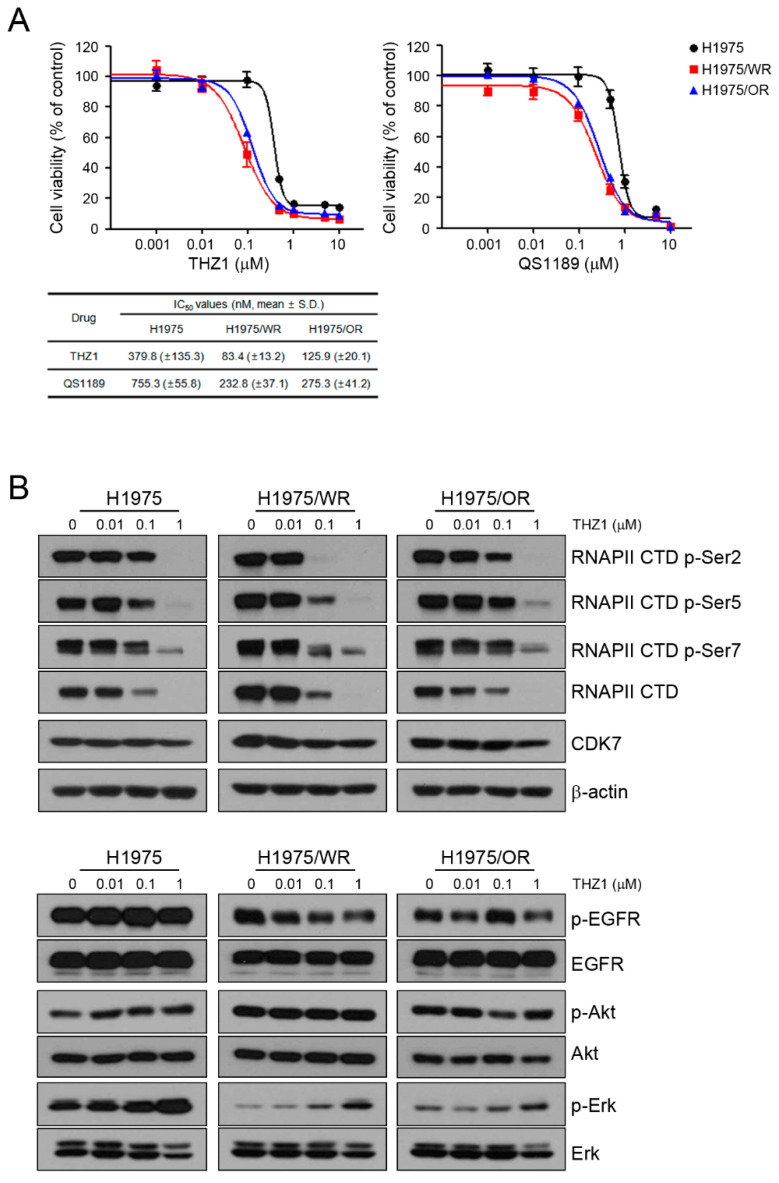
Effects of CDK7 inhibitors on cells with acquired resistance to WZ4002 or osimertinib. (**A**) Cells were treated with the indicated doses of THZ1 or QS1189 for 72 h, and cell viability was determined using MTT assays. The IC50 values of the CDK7 inhibitors were calculated. (**B**) Cells were treated with the indicated doses of THZ1 for 6 h. The indicated protein levels were analyzed by western blotting.

**Figure 3 cells-09-02596-f003:**
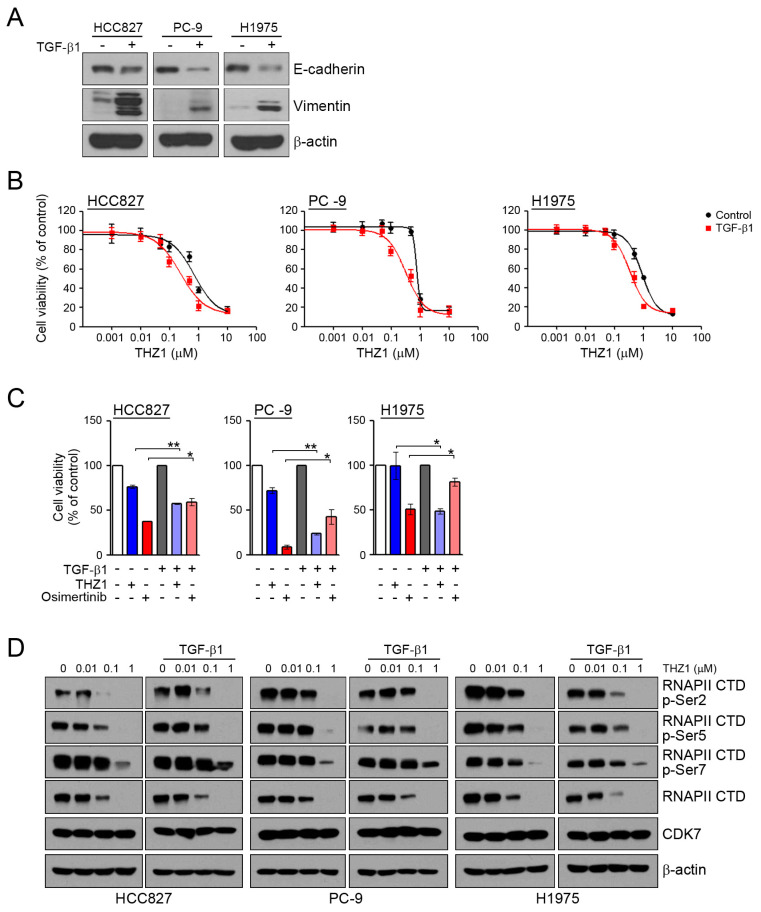
Effects of the CDK7 inhibitors on TGF-β1 stimulated EMT. (**A**) Cells were treated with TGF-β1 (10 ng/mL) for 24 h, and the levels of EMT-related proteins were analyzed by western blotting. (**B**) Cells were pretreated with TGF-β1 (10 ng/mL) for 24 h and then incubated with the indicated doses of THZ1 for 72 h, and cell viability was determined by an MTT assay. (**C**) Cells were pretreated with TGF-β1 for 24 h and then incubated with 0.1 μM THZ1 or 0.1 μM osimertinib for 48 h. Cell viability was determined by cell counting. (**D**) Cells were treated with the indicated doses of THZ1 in the presence or absence of TGF-β1 for 6 h. The indicated protein levels were analyzed by western blotting. * *P* < 0.05; ** *P* < 0.005.

**Figure 4 cells-09-02596-f004:**
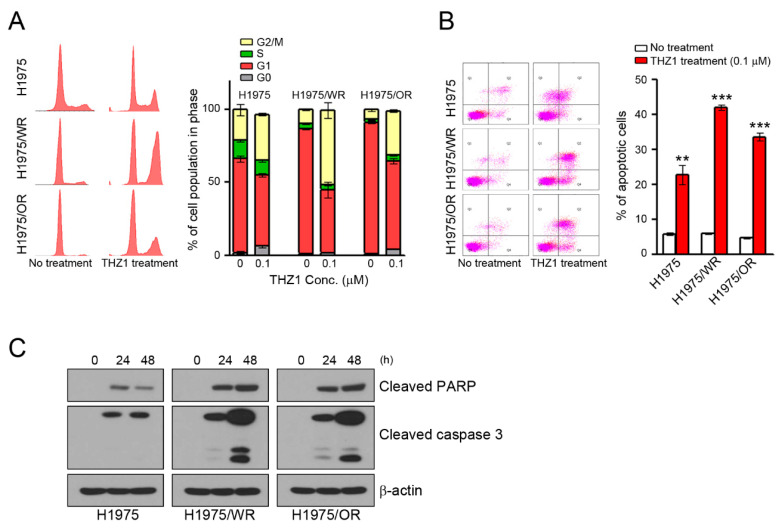
Induction of cell cycle arrest and apoptosis by CDK7 inhibitors in parental and resistant cells. (**A**,**B**) Cells were treated with THZ1 for 24 or 48 h to analyze the cell cycle and apoptosis. The cell cycle was determined by propidium iodide (PI) staining, and apoptosis was determined by Annexin V-FITC and PI. The experiment was conducted in triplicate. (**C**) Cells were treated with 0.1 μM THZ1 in a time-dependent manner. Cleaved PARP and caspase-3 were detected by western blotting. ** *P* < 0.005; *** *P* < 0.0005 compared to the control group.

**Figure 5 cells-09-02596-f005:**
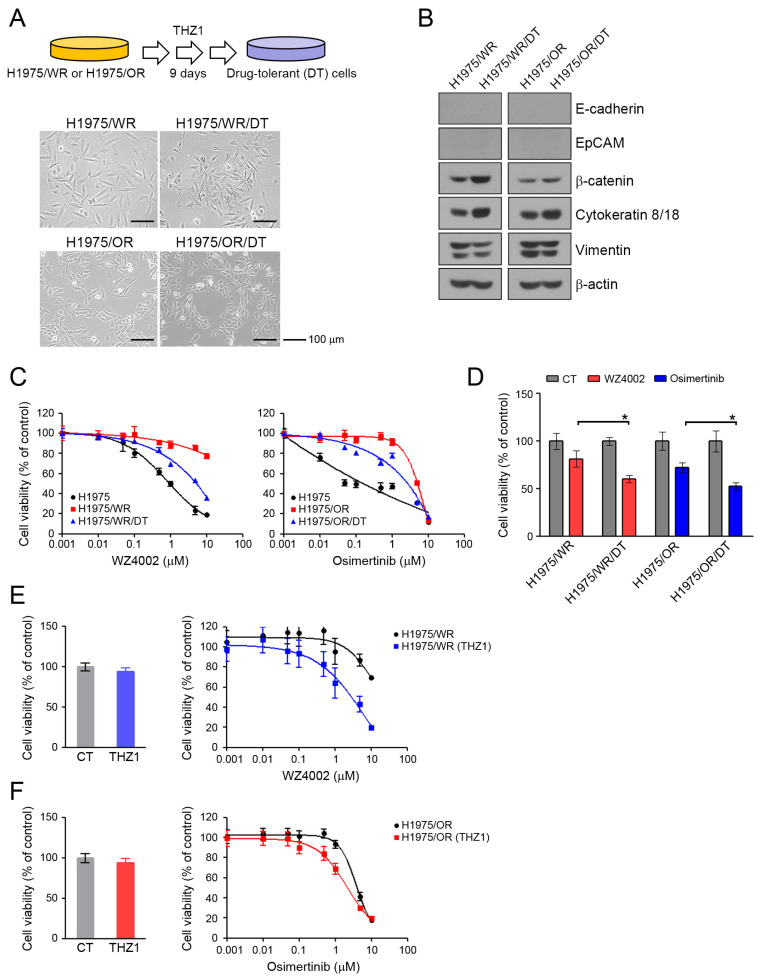
Morphological changes and recovery of the sensitivity to EGFR-TKIs by THZ1-tolerant cells. (**A**) DT cells were generated by repeated exposure to 100 nM THZ1 (3 times in 9 days). Morphologic changes were determined by using a light microscope. (**B**) EMT-related molecules were analyzed by western blotting. (**C**) Cells were treated with the indicated doses of WZ4002 or osimertinib for 72 h, and cell viability was determined by the MTT assay. (**D**) Cells were treated with 1 μM WZ4002 or osimertinib for 48 h. Cell viability was determined by cell counting. (**E**,**F**) Cells were treated with the indicated doses of a single drug (WZ4002 or osimertinib) or in combination with 10 nM THZ1 for 72 h. Cell viability was determined by the MTT assay. * *P* < 0.05.

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
