# Peer review of "Efficacy of the CDK7 Inhibitor on EMT-Associated Resistance to 3rd Generation EGFR-TKIs in Non-Small Cell Lung Cancer Cell Lines"

_cells, 2020, doi:10.3390/cells9122596_

Round 1

Reviewer 1 Report

I only ask to check the text where the IC50 values for THZ1 of DT cells are reported, because there are some errors.

In addition, I could not find the number of experiments performed fo the WB epxeriments (considering that those shown are representative).

Author Response

I only ask to check the text where the IC50 values for THZ1 of DT cells are reported, because there are some errors.

Our reply: We corrected them as requested by the reviewer in the revised manuscript.

In addition, I could not find the number of experiments performed fo the WB epxeriments (considering that those shown are representative).

Our reply: we confirmed the WB experiments on our manuscript. However, we can not find the loss or missing of WB experiments. If reviewer indicate no band of E-cadherin and EpCAM in the figure 5B, E-cadherin and EpCAM in resistant cells did not detected in the figure 1C. Although our manuscript did not demonstrate the mechanisms of loss of E-cadherin and EpCAM, they did not detected for long time-exposures in the figure 5B.

Reviewer 2 Report

The authors addressed all questions and improved the quality of data in the revised manuscript.

Author Response

The authors addressed all questions and improved the quality of data in the revised manuscript.

Our reply: we appreciate the thoughtful feedback and suggestions, as well as the positive comments, on our manuscript.

Reviewer 3 Report

Efficacy of the CDK7 inhibitor on EMT-associated resistance to 3rd generation EGFR-TKIs in non-small cell lung cancer lines

Thank you for giving me the opportunity to review this manuscript. I find it interesting, well written and clinical relevant. I have a few comments/questions that I think the authors should answer before it publication.

Line 91: please indicate the EGFR mutational status for the used cell lines.

Line 91-96: please describe more detailed how the resistant cell lines were established e.g. drug concentrations, incubation times and number of passages.

Line 91-96: please describe the drug WZ4002 and why was this drug used?

Line 111-119: which antibody dilutions were used in the immunoblotting experiments?

Line 145-162: is the H1975/OR cell line also resistant to first and second generation anti-EGFR TKI drugs?

Discussion: can new secondary EGFR mutations be excluded as the resistance mechanism of resistant cell lines?

Author Response

We attached our reply to reviewer.

This manuscript is a resubmission of an earlier submission. The following is a list of the peer review reports and author responses from that submission.

Round 1

Reviewer 1 Report

Ji et al. show that EMT is associated with acquired resistance to 3rd generation EGFR-TKI and CDK7 inhibitor, THZ1 induces MET in the resistant NSCLC cells. Also, THZ1-tolerant cells recover the sensitivity to 3rd generation EGFR-TKI. Overall, the authors demonstrated the correlation between EMT and acquired resistance to 3rd generation EGFR-TKI and then suggest a potential therapeutic strategy to overcome the RGFR-TKI resistance.

The study's interesting finding enriches the understanding of the resistance mechanism of 3rd generation EGFR-TKI in NSCLC. However, there are minor questions that should be addressed or corrected as specific comments to be published.

  1. The authors established the 3rd generation RGFR-TKI resistant cells from H1975 cell line. The resistant cells showed apparent phenotypes related to EMT and the resistance. However, the phenotypes of the resistant cells could be unique features of H1975 cell line. The other resistant cells from other NSCLC cell lines can strongly support the finding in the study. 

  1. In Figure 5A, it is difficult to recognize the morphological change due to the low magnitude images. The photos should be replaced with a higher magnitude image.

  1. In Figure 5B, the EMT associated markers showed a minor change in the THZ1 tolerant cells. In particular, the epithelial markers (i.e. E-cadherin, EpCAM) were not detected. The authors should provide more explanation in the Discussion section. 

  1. The quantitation of beta-catenin, CK8/18, and vimentin would help recognize the change of EMT markers in Figure 5B. 

Author Response

We added the response to the reviewer's comments as attached file.

Reviewer 2 Report

In the manuscript “Efficacy of the CDK7 Inhibitor on EMT-Associated Resistance to 3rd Generation EGFR-TKIs in Non-Small Cell Lung Cancer”, Ji eta al. demonstrate that CDK7 inhibitors can overcome EMT-associated resistance to EGFR-TKIs in NSCLC cell models.

The manuscript is potentially interesting, although some issues need to be addressed.

Major points:

The molecular characterization of the resistant cells is not complete. The authors state that the experiments were performed after >7 week of drug removal. In Figure 1C, they should show the effects on the molecular pathways also in the presence of the EGFR inhibitors to clarify why EGFR activation/expression and p-ERK are reduced in the resistant cells, while p-AKT is up-regulated.

Figure 3: The authors should demonstrate that EMT is effectively induced after 24 hours of treatment with TGF-β1 by showing the changes in the expression of epithelial and mesenchymal markers.

Since THZ1 inhibits RNAPII CTD in both the sensitive and resistant cells as well as in TGF-β1-treated cells, it is not clear which is the mechanism underlying the higher sensitivity of EMT-induced cells. The authors state that this is one of the limitation of their study and simply report, in a supplementary figure, that myc and CITED2 proteins are overexpressed in the resistant cells. This figure should be moved to the Results Section. In addition, in my opinion, the interest of the manuscript would increase greatly if the authors analyzed the involvement of these proteins or other regulatory pathways in EMT and sensitivity to CDK7 inhibitors.

The histograms in Figure S3 are not clear. Indeed, it seems that treatment with 100nM THZ1 is ineffective in EGFR-TKIs resistant cells, although their IC50 is around 83 and 125nM for WZ4002 and Osimertinib, respectively. The results of Figure S3 should be reported and commented in the Results Section.

The authors should report the IC50 value for THZ1 of DT cells generated from H1975/WR and H1975/OR, to have a comparison with H1975 parental cells. Indeed, it is not clear whether H1975 cells can be considered sensitive or resistant to THZ1.

Minor points:

  • Please, change the title to “Efficacy of CDK7 inhibitors on [….] in Non-Small Cell Lung Cancer cell lines”. It is more correct to specify that the study was performed only in in vitro cell models.

  • In the Introduction Section, the authors should give more information about the THZ1 compound.

  • The material and methods section should be more accurate. Some examples:

Line 83: the information about the cell medium used is missing.

Immunoblotting: some antibodies are not indicated. Please carefully check that all the antibodies used are listed.

Line 123: The invasion and not the migration assay is here described.

  • The number of the experiments performed is not always clearly indicated.

Author Response

We added the response to the reviewer's comments as attached files.

Round 2

Reviewer 2 Report

  1. The histograms in Figure S3 are not clear. Indeed, it seems that treatment with 100nM THZ1 is ineffective in EGFR-TKIs resistant cells, although their IC50 is around 83 and 125nM for WZ4002 and Osimertinib, respectively. The results of Figure S3 should be reported and commented in the Results Section.

Our reply: We found that the dose of THZ1 is incorrectly listed in Figure S3. We corrected the dose of THZ1 (10 nM) in the revised manuscript. In addition, we added text describing the results of Figure S3 to the results section.

New comment: Considering that 100nM is the concentration used in all the other experiments, it is not clear why the authors used 10nM, which is a concentration that has no effect on RNAPII CTD phosphorylation. In my opinion, it would be more correct to repeat the experiment using 100nM.

  1. The authors should report the IC50 value for THZ1 of DT cells generated from H1975/WR and H1975/OR, to have a comparison with H1975 parental cells. Indeed, it is not clear whether H1975 cells can be considered sensitive or resistant to THZ1.

Our reply: We commented on the IC50 values as requested by the reviewer in the results section.

New comment: I asked to show the IC50 values for THZ1 of DT cells and not for EGFRTKIs.

  1. Line 123: The invasion and not the migration assay is here described.

Our reply: We have added the requested information and made corrections accordingly and we thank the reviewer for pointing this out.

New comment: Line 125. Sentence: “The migration assays used the same procedure with filters that were coated with extracellular matrix on their upper surface”. Change migration with invasion.

  1. The number of the experiments performed is not always clearly indicated

Our reply: We have made corrections accordingly and thank the reviewer for pointing this out.

New comment: The number of the experiments performed is still missing for some experiments (for example, western blotting experiments). Please check carefully and add the required information.